# Offline Reinforcement Learning as One Big Sequence Modeling Problem

**Michael Janner**     **Qiyang Li**     **Sergey Levine**
University of California at Berkeley
{janner, qcli}@berkeley.edu     svlevine@eecs.berkeley.edu

## Abstract

Reinforcement learning (RL) is typically concerned with estimating stationary policies or single-step models, leveraging the Markov property to factorize problems in time. However, we can also view RL as a generic sequence modeling problem, with the goal being to produce a sequence of actions that leads to a sequence of high rewards. Viewed in this way, it is tempting to consider whether high-capacity sequence prediction models that work well in other domains, such as natural-language processing, can also provide effective solutions to the RL problem. To this end, we explore how RL can be tackled with the tools of sequence modeling, using a Transformer architecture to model distributions over trajectories and repurposing beam search as a planning algorithm. Framing RL as sequence modeling problem simplifies a range of design decisions, allowing us to dispense with many of the components common in offline RL algorithms. We demonstrate the flexibility of this approach across long-horizon dynamics prediction, imitation learning, goal-conditioned RL, and offline RL. Further, we show that this approach can be combined with existing model-free algorithms to yield a state-of-the-art planner in sparse-reward, long-horizon tasks.

## 1   Introduction

The standard treatment of reinforcement learning relies on decomposing a long-horizon problem into smaller, more local subproblems. In model-free algorithms, this takes the form of the principle of optimality (Bellman, 1957), a recursion that leads naturally to the class of dynamic programming methods like $Q$-learning. In model-based algorithms, this decomposition takes the form of single-step predictive models, which reduce the problem of predicting high-dimensional, policy-dependent state trajectories to that of estimating a comparatively simpler, policy-agnostic transition distribution.

However, we can also view reinforcement learning as analogous to a sequence generation problem, with the goal being to produce a sequence of actions that, when enacted in an environment, will yield a sequence of high rewards. In this paper, we consider the logical extreme of this analogy: does the toolbox of contemporary sequence modeling itself provide a viable reinforcement learning algorithm? We investigate this question by treating trajectories as unstructured sequences of states, actions, and rewards. We model the distribution of these trajectories using a Transformer architecture (Vaswani et al., 2017), the current tool of choice for capturing long-horizon dependencies. In place of the trajectory optimizers common in model-based control, we use beam search (Reddy, 1977), a heuristic decoding scheme ubiquitous in natural language processing, as a planning algorithm.

Posing reinforcement learning, and more broadly data-driven control, as a sequence modeling problem handles many of the considerations that typically require distinct solutions: actor-critic algorithms require separate actors and critics, model-based algorithms require predictive dynamics models, and offline RL methods often require estimation of the behavior policy (Fujimoto et al., 2019). These

---

Code is available at `trajectory-transformer.github.io`

35th Conference on Neural Information Processing Systems (NeurIPS 2021).

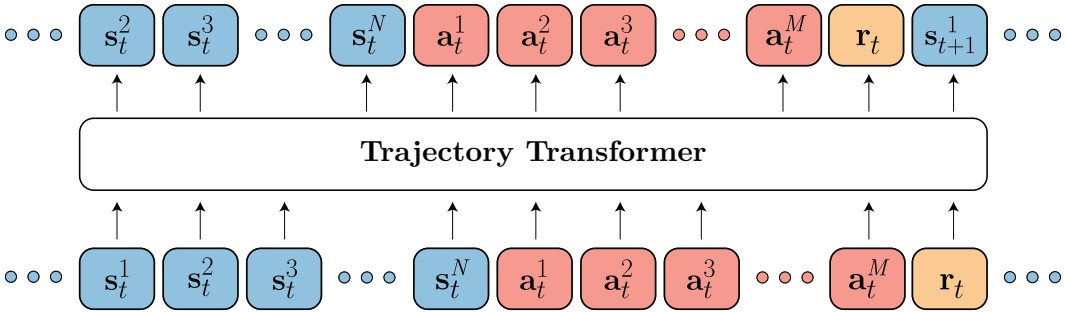

**Figure 1 (Architecture)** The Trajectory Transformer trains on sequences of (autoregressively discretized) states, actions, and rewards. Planning with the Trajectory Transformer mirrors the sampling procedure used to generate sequences from a language model.

components estimate different densities or distributions, such as that over actions in the case of actors and behavior policies, or that over states in the case of dynamics models. Even value functions can be viewed as performing inference in a graphical model with auxiliary optimality variables, amounting to estimation of the distribution over future rewards (Levine, 2018). All of these problems can be unified under a single sequence model, which treats states, actions, and rewards as simply a stream of data. The advantage of this perspective is that high-capacity sequence model architectures can be brought to bear on the problem, resulting in a more streamlined approach that could benefit from the same scalability underlying large-scale unsupervised learning results (Brown et al., 2020).

We refer to our model as a Trajectory Transformer (Figure 1) and evaluate it in the offline regime so as to be able to make use of large amounts of prior interaction data. The Trajectory Transformer is a substantially more reliable long-horizon predictor than conventional dynamics models, even in Markovian environments for which the standard model parameterization is in principle sufficient. When decoded with a modified beam search procedure that biases trajectory samples according to their cumulative reward, the Trajectory Transformer attains results on offline RL benchmarks that are competitive with the best prior methods designed specifically for that setting. Additionally, we describe how variations of the same decoding procedure yield a model-based imitation learning method, a goal-reaching method, and, when combined with dynamic programming, a state-of-the-art planner for sparse-reward, long-horizon tasks. Our results suggest that the algorithms and architectural motifs that have been widely applicable in unsupervised learning carry similar benefits in RL.

## 2   Related Work

Recent advances in sequence modeling with deep networks have led to rapid improvement in the effectiveness of such models, from LSTMs and sequence-to-sequence models (Hochreiter & Schmidhuber, 1997; Sutskever et al., 2014) to Transformer architectures with self-attention (Vaswani et al., 2017). In light of this, it is tempting to consider how such sequence models can lead to improved performance in RL, which is also concerned with sequential processes (Sutton, 1988). Indeed, a number of prior works have studied applying sequence models of various types to represent components in standard RL algorithms, such as policies, value functions, and models (Bakker, 2002; Heess et al., 2015a; Chiappa et al., 2017; Parisotto et al., 2020; Parisotto & Salakhutdinov, 2021; Kumar et al., 2020b). While such works demonstrate the importance of such models for representing memory (Oh et al., 2016), they still rely on standard RL algorithmic advances to improve performance. The goal in our work is different: we aim to replace as much of the RL pipeline as possible with sequence modeling, so as to produce a simpler method whose effectiveness is determined by the representational capacity of the sequence model rather than algorithmic sophistication.

Estimation of probability distributions and densities arises in many places in learning-based control. This is most obvious in model-based RL, where it is used to train predictive models that can then be used for planning or policy learning (Sutton, 1990; Silver et al., 2008; Fairbank, 2008; Deisenroth & Rasmussen, 2011; Lampe & Riedmiller, 2014; Heess et al., 2015b; Janner et al., 2020; Amos et al., 2021). However, it also figures heavily in offline RL, where it is used to estimate conditional distributions over actions that serve to constrain the learned policy to avoid out-of-

distribution behavior that is not supported under the dataset (Fujimoto et al., 2019; Kumar et al., 2019a; Ghasemipour et al., 2021); imitation learning, where it is used to fit an expert's actions to obtain a policy (Ross & Bagnell, 2010; Ross et al., 2011); and other areas such as hierarchical RL (Peng et al., 2017; Co-Reyes et al., 2018; Jiang et al., 2019). In our method, we train a single high-capacity sequence model to represent the joint distribution over sequences of states, actions, and rewards. This serves as *both* a predictive model *and* a behavior policy (for imitation) or behavior constraint (for offline RL).

Our approach to RL is most closely related to prior model-based methods that plan with a learned model (Chua et al., 2018; Wang & Ba, 2020). However, while these prior methods typically require additional machinery to work well, such as ensembles in the online setting (Kurutach et al., 2018; Buckman et al., 2018; Malik et al., 2019) or conservatism mechanisms in the offline setting (Yu et al., 2020; Kidambi et al., 2020; Argenson & Dulac-Arnold, 2021), our method does not require explicit handling of these components. Modeling the states and actions jointly already provides a bias toward generating in-distribution actions, which avoids the need for explicit pessimism (Fujimoto et al., 2019; Kumar et al., 2019a; Ghasemipour et al., 2021; Nair et al., 2020; Jin et al., 2021; Yin et al., 2021; Dadashi et al., 2021). Our method also differs from most prior model-based algorithms in the dynamics model architecture used, with fully-connected networks parameterizing diagonal-covariance Gaussian distributions being a common choice (Chua et al., 2018), though recent work has highlighted the effectiveness of autoregressive state prediction (Zhang et al., 2021) like that used by the Trajectory Transformer. In the context of recently proposed offline RL algorithms, our method can be interpreted as a combination of model-based RL and policy constraints (Kumar et al., 2019a; Wu et al., 2019), though our approach does not require introducing such constraints explicitly. In the context of model-free RL, our method also resembles recently proposed work on goal relabeling (Andrychowicz et al., 2017; Rauber et al., 2019; Ghosh et al., 2021; Paster et al., 2021) and reward conditioning (Schmidhuber, 2019; Srivastava et al., 2019; Kumar et al., 2019b) to reinterpret all past experience as useful demonstrations with proper contextualization.

Concurrently with our work, Chen et al. (2021) also proposed an RL approach centered around sequence prediction, focusing on reward conditioning as opposed to the beam-search-based planning used by the Trajectory Transformer. Their work further supports the possibility that a high-capacity sequence model can be applied to reinforcement learning problems without the need for the components usually associated with RL algorithms.

# 3 Reinforcement Learning and Control as Sequence Modeling

In this section, we describe the training procedure for our sequence model and discuss how it can be used for control. We refer to the model as a Trajectory Transformer for brevity, but emphasize that at the implementation level, both our model and search strategy are nearly identical to those common in natural language processing. As a result, modeling considerations are concerned less with architecture design and more with how to represent trajectory data – potentially consisting of continuous states and actions – for processing by a discrete-token architecture (Radford et al., 2018).

## 3.1 Trajectory Transformer

At the core of our approach is the treatment of trajectory data as an unstructured sequence for modeling by a Transformer architecture. A trajectory $\tau$ consists of $T$ states, actions, and scalar rewards:

$$\tau = \left(\mathbf{s}_1, \mathbf{a}_1, r_1, \mathbf{s}_2, \mathbf{a}_2, r_2, \ldots, \mathbf{s}_T, \mathbf{a}_T, r_T\right).$$

In the event of continuous states and actions, we discretize each dimension independently. Assuming $N$-dimensional states and $M$-dimensional actions, this turns $\tau$ into sequence of length $T(N+M+1)$:

$$\tau = \left(\ldots, s_t^1, s_t^2, \ldots, s_t^N, a_t^1, a_t^2, \ldots, a_t^M, r_t, \ldots\right) \qquad t = 1, \ldots, T.$$

Subscripts on all tokens denote timestep and superscripts on states and actions denote dimension (*i.e.*, $s_t^i$ is the $i^{\text{th}}$ dimension of the state at time $t$). While this choice may seem inefficient, it allows us to model the distribution over trajectories with more expressivity without simplifying assumptions such as Gaussian transitions.

---
**Algorithm 1** Beam search
---
1: **Require** Input sequence $\mathbf{x}$, vocabulary $\mathcal{V}$, sequence length $T$, beam width $B$
2: **Initialize** $Y_0 = \{\,(\,)\,\}$
3: **for** $t = 1, \ldots, T$ **do**
4:     $\mathcal{C}_t \leftarrow \{\mathbf{y}_{t-1} \circ y \mid \mathbf{y}_{t-1} \in Y_{t-1} \text{ and } y \in \mathcal{V}\}$       // candidate single-token extensions
5:     $Y_t \leftarrow \underset{Y \subseteq \mathcal{C}_t,\, |Y|=B}{\operatorname{argmax}} \log P_\theta(Y \mid \mathbf{x})$       // $B$ most likely sequences from candidates
6: **end for**
7: **Return** $\underset{\mathbf{y} \in Y_T}{\operatorname{argmax}} \log P_\theta(\mathbf{y} \mid \mathbf{x})$
---

We investigate two simple discretization approaches:

1. **Uniform:** All tokens for a given dimension correspond to a fixed width of the original continuous space. Assuming a per-dimension vocabulary size of $V$, the tokens for state dimension $i$ cover uniformly-spaced intervals of width $(\max \mathbf{s}^i - \min \mathbf{s}^i)/V$.

2. **Quantile:** All tokens for a given dimension account for an equal amount of probability mass under the empirical data distribution; each token accounts for 1 out of every $V$ data points in the training set.

Uniform discretization has the advantage that it retains information about Euclidean distance in the original continuous space, which may be more reflective of the structure of a problem than the training data distribution. However, outliers in the data may have outsize effects on the discretization size, leaving many tokens corresponding to zero training points. The quantile discretization scheme ensures that all tokens are represented in the data. We compare the two empirically in Section 4.2.

Our model is a Transformer decoder mirroring the GPT architecture (Radford et al., 2018). We use a smaller architecture than those typically used in large-scale language modeling, consisting of four layers and four self-attention heads. (A full architectural description is provided in Appendix **??**.) Training is performed with the standard teacher-forcing procedure (Williams & Zipser, 1989) used to train sequence models. Denoting the parameters of the Trajectory Transformer as $\theta$ and induced conditional probabilities as $P_\theta$, the objective maximized during training is:

$$\mathcal{L}(\tau) = \sum_{t=1}^{T} \Big( \sum_{i=1}^{N} \log P_\theta\big(s_t^i \mid \mathbf{s}_t^{<i}, \boldsymbol{\tau}_{<t}\big) + \sum_{j=1}^{M} \log P_\theta\big(a_t^j \mid \mathbf{a}_t^{<j}, \mathbf{s}_t, \boldsymbol{\tau}_{<t}\big) + \log P_\theta\big(r_t \mid \mathbf{a}_t, \mathbf{s}_t, \boldsymbol{\tau}_{<t}\big) \Big),$$

in which we use $\boldsymbol{\tau}_{<t}$ to denote a trajectory from timesteps 0 through $t-1$, $\mathbf{s}_t^{<i}$ to denote dimensions 0 through $i-1$ of the state at timestep $t$, and similarly for $\mathbf{a}_t^{<j}$. We use the Adam optimizer (Kingma & Ba, 2015) with a learning rate of $2.5 \times 10^{-4}$ to train parameters $\theta$.

## 3.2  Planning with Beam Search

We now describe how sequence generation with the Trajectory Transformer can be repurposed for control, focusing on three settings: imitation learning, goal-conditioned reinforcement learning, and offline reinforcement learning. These settings are listed in increasing amount of required modification on top of the sequence model decoding approach routinely used in natural language processing.

The core algorithm providing the foundation of our planning techniques, beam search, is described in Algorithm 1 for generic sequences. Following the presentation in Meister et al. (2020), we have overloaded $\log P_\theta(\cdot \mid \mathbf{x})$ to define the likelihood of a set of sequences in addition to that of a single sequence: $\log P_\theta(Y \mid x) = \sum_{\mathbf{y} \in Y} \log P_\theta(\mathbf{y} \mid \mathbf{x})$. We use $(\,)$ to denote the empty sequence and $\circ$ to represent concatenation.

**Imitation learning.**   When the goal is to reproduce the distribution of trajectories in the training data, we can optimize directly for the probability of a trajectory $\boldsymbol{\tau}$. This situation matches the goal of sequence modeling exactly and as such we may use Algorithm 1 without modification by setting the conditioning input $\mathbf{x}$ to the current state $\mathbf{s}_t$ (and optionally previous history $\boldsymbol{\tau}_{<t}$).

The result of this procedure is a tokenized trajectory $\boldsymbol{\tau}$, beginning from a current state $\mathbf{s}_t$, that has high probability under the data distribution. If the first action $\mathbf{a}_t$ in the sequence is enacted and beam

search is repeated, we have a receding horizon-controller. This approach resembles a long-horizon model-based variant of behavior cloning, in which entire trajectories are optimized to match those of a reference behavior instead of only immediate state-conditioned actions. If we set the predicted sequence length to be the action dimension, our approach corresponds exactly to the simplest form of behavior cloning with an autoregressive policy.

**Goal-conditioned reinforcement learning.**    Transformer architectures feature a "causal" attention mask to ensure that predictions only depend on previous tokens in a sequence. In the context of natural language, this design corresponds to generating sentences in the linear order in which they are spoken as opposed to an ordering reflecting their hierarchical syntactic structure (see, however, Gu et al. 2019 for a discussion of non-left-to-right sentence generation with autoregressive models). In the context of trajectory prediction, this choice instead reflects physical causality, disallowing future events to affect the past. However, the conditional probabilities of the past given the future are still well-defined, allowing us to condition samples not only on the preceding states, actions, and rewards that have already been observed, but also any future context that we wish to occur. If the future context is a state at the end of a trajectory, we decode trajectories with probabilities of the form:

$$P_\theta(s_t^i \mid \mathbf{s}_t^{<i}, \boldsymbol{\tau}_{<t}, \mathbf{s}_T)$$

We can use this directly as a goal-reaching method by conditioning on a desired final state $\mathbf{s}_T$. If we always condition sequences on a final goal state, we may leave the lower-diagonal attention mask intact and simply permute the input trajectory to $\{\mathbf{s}_T, \mathbf{s}_1, \mathbf{s}_2, \ldots, \mathbf{s}_{T-1}\}$. By prepending the goal state to the beginning of a sequence, we ensure that all other predictions may attend to it without modifying the standard attention implementation. This procedure for conditioning resembles prior methods that use supervised learning to train goal-conditioned policies (Ghosh et al., 2021) and is also related to relabeling techniques in model-free RL (Andrychowicz et al., 2017). In our framework, it is identical to the standard subroutine in sequence modeling: inferring the most likely sequence given available evidence.

**Offline reinforcement learning.**    The beam search method described in Algorithm 1 optimizes sequences for their probability under the data distribution. By replacing the log-probabilities of transitions with the predicted reward signal, we can use the same Trajectory Transformer and search strategy for reward-maximizing behavior. Appealing to the control as inference graphical model (Levine, 2018), we are in effect replacing a transition's log-probability in beam search with its log-probability of optimality.

Using beam-search as a reward-maximizing procedure has the risk of leading to myopic behavior. To address this issue, we augment each transition in the training trajectories with reward-to-go: $R_t = \sum_{t'=t}^{T} \gamma^{t'-t} r_{t'}$ and include it as an additional quantity, discretized identically to the others, to be predicted after immediate rewards $r_t$. During planning, we then have access to value estimates from our model to add to cumulative rewards. While acting greedily with respect to such Monte Carlo value estimates is known to suffer from poor sample complexity and convergence to suboptimal behavior when online data collection is not allowed, we only use this reward-to-go estimate as a heuristic to guide beam search, and hence our method does not require the estimated values to be as accurate as in methods that rely solely on the value estimates to select actions.

In offline RL, reward-to-go estimates are functions of the *behavior* policy that collected the training data and do not, in general, correspond to the values achieved by the Trajectory Transformer-derived policy. Of course, it is much simpler to learn the value function of the behavior policy than that of the optimal policy, since we can simply use Monte Carlo estimates without relying on Bellman updates. A value function for an improved policy would provide a better search heuristic, though requires invoking the tools of dynamic programming. In Section 4.2 we show that the simple reward-to-go estimates are sufficient for planning with the Trajectory Transformer in many environments, but that improved value functions are useful in the most challenging settings, such as sparse-reward tasks.

Because the Trajectory Transformer predicts reward and reward-to-go only every $N + M + 1$ tokens, we sample all intermediate tokens according to model log-probabilities, as in the imitation learning and goal-reaching settings. More specifically, we sample full transitions $(\mathbf{s}_t, \mathbf{a}_t, r_t, R_t)$ using likelihood-maximizing beam search, treat these transitions as our vocabulary, and filter sequences of transitions by those with the highest cumulative reward plus reward-to-go estimate.

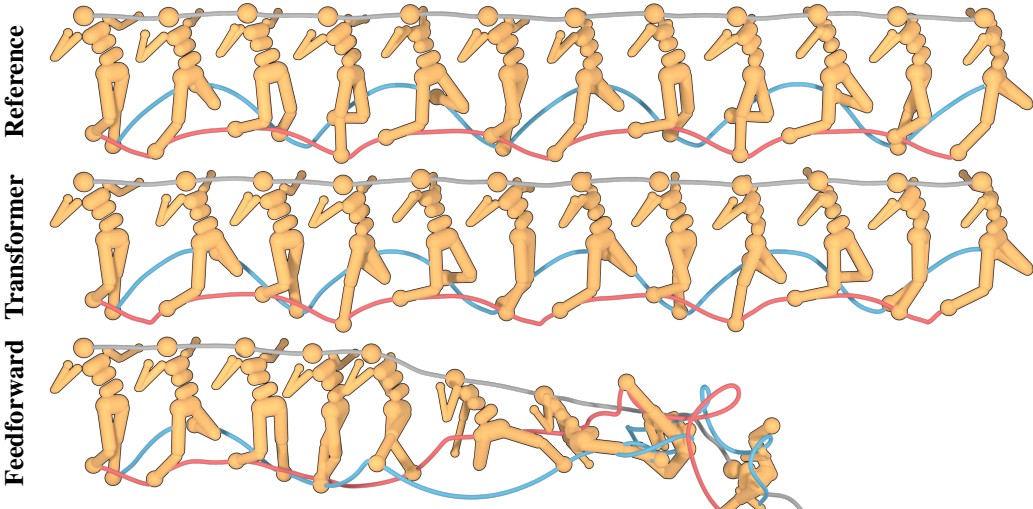

**Figure 2 (Prediction visualization)** A qualitative comparison of length-100 trajectories generated by the Trajectory Transformer and a feedforward Gaussian dynamics model from PETS, a state-of-the-art planning algorithm Chua et al. (2018). Both models were trained on trajectories collected by a single policy, for which a true trajectory is shown for reference. Compounding errors in the single-step model lead to physically implausible predictions, whereas the Transformer-generated trajectory is visually indistinguishable from those produced by the policy acting in the actual environment. The paths of the feet and head are traced through space for depiction of the movement between rendered frames.

We have taken a sequence-modeling route to what could be described as a fairly simple-looking model-based planning algorithm, in that we sample candidate action sequences, evaluate their effects using a predictive model, and select the reward-maximizing trajectory. This conclusion is in part due to the close relation between sequence modeling and trajectory optimization. There is one dissimilarity, however, that is worth highlighting: by modeling actions jointly with states and sampling them using the same procedure, we can prevent the model from being queried on out-of-distribution actions. The alternative, of treating action sequences as unconstrained optimization variables that do not depend on state (Nagabandi et al., 2018), can more readily lead to model exploitation, as the problem of maximizing reward under a learned model closely resembles that of finding adversarial examples for a classifier (Goodfellow et al., 2014).

## 4 Experiments

Our experimental evaluation focuses on (1) the accuracy of the Trajectory Transformer as a long-horizon predictor compared to standard dynamics model parameterizations and (2) the utility of sequence modeling tools – namely beam search – as a control algorithm in the context of offline reinforcement learning, imitation learning, and goal-reaching.

### 4.1 Model Analysis

We begin by evaluating the Trajectory Transformer as a long-horizon policy-conditioned predictive model. The usual strategy for predicting trajectories given a policy is to rollout with a single-step model, with actions supplied by the policy. Our protocol differs from the standard approach not only in that the model is not Markovian, but also in that it does not require access to a policy to make predictions – the outputs of the policy are modeled alongside the states encountered by that policy. Here, we focus only on the quality of the model's predictions; we use actions predicted by the model for an imitation learning method in the next subsection.

**Trajectory predictions.** Figure 2 depicts a visualization of predicted 100-timestep trajectories from our model after having trained on a dataset collected by a trained humanoid policy. Though model-based methods have been applied to the humanoid task, prior works tend to keep the horizon

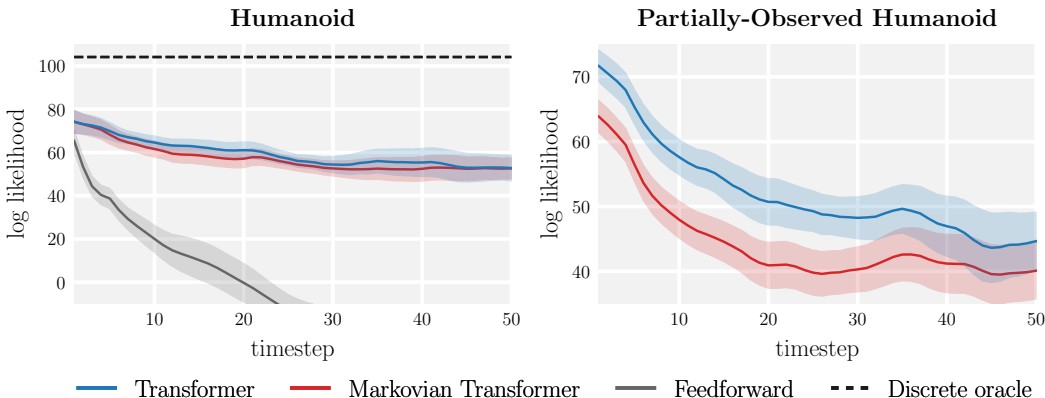

**Figure 3 (Compounding model errors)** We compare the accuracy of the Trajectory Transformer (with uniform discretization) to that of the probabilistic feedforward model ensemble (Chua et al., 2018) over the course of a planning horizon in the humanoid environment, corresponding to the trajectories visualized in Figure 2. The Trajectory Transformer has substantially better error compounding with respect to prediction horizon than the feedforward model. The discrete oracle is the maximum log likelihood attainable given the discretization size; see Appendix **??** for a discussion.

intentionally short to prevent the accumulation of model errors (Janner et al., 2019; Amos et al., 2021). The reference model is the probabilistic ensemble implementation of PETS (Chua et al., 2018); we tuned the number of models within the ensemble, the number of layers, and layer sizes, but were unable to produce a model that predicted accurate sequences for more than a few dozen steps. In contrast, we see that the Trajectory Transformer's long-horizon predictions are substantially more accurate, remaining visually indistinguishable from the ground-truth trajectories even after 100 predicted steps. To our knowledge, no prior model-based RL algorithm has demonstrated predicted rollouts of such accuracy and length on tasks of comparable dimensionality.

**Error accumulation.**    A quantitative account of the same finding is provided in Figure 3, in which we evaluate the model's accumulated error versus prediction horizon. Standard predictive models tend to have excellent single-step errors but poor long-horizon accuracy, so instead of evaluating a test-set single-step likelihood, we sample 1000 trajectories from a fixed starting point to estimate the per-timestep state marginal predicted by each model. We then report the likelihood of the states visited by the reference policy on a held-out set of trajectories under these predicted marginals. To evaluate the likelihood under our discretized model, we treat each bin as a uniform distribution over its specified range; by construction, the model assigns zero probability outside of this range.

To better isolate the source of the Transformer's improved accuracy over standard single-step models, we also evaluate a Markovian variant of our same architecture. This ablation has a truncated context window that prevents it from attending to more than one timestep in the past. This model performs similarly to the trajectory Transformer on fully-observed environments, suggesting that architecture differences and increased expressivity from the autoregressive state discretization play a large role in the trajectory Transformer's long-horizon accuracy. We construct a partially-observed version of the same humanoid environment, in which each dimension of every state is masked out with 50% probability (Figure 3 right), and find that, as expected, the long-horizon conditioning plays a larger role in the model's accuracy in this setting.

**Attention patterns.**    We visualize the attention maps during model predictions in Figure 4. We find two primary attention patterns. The first is a discovered Markovian strategy, in which a state prediction attends overwhelmingly to the previous transition. The second is qualitatively striated, with the model attending to specific dimensions in multiple prior states for each state prediction. Simultaneously, the action predictions attend to prior actions more than they do prior states. The action dependencies contrast with the usual formulation of behavior cloning, in which actions are a function of only past states, but is reminiscent of the action filtering technique used in some planning algorithm to produce smoother action sequences (Nagabandi et al., 2019).

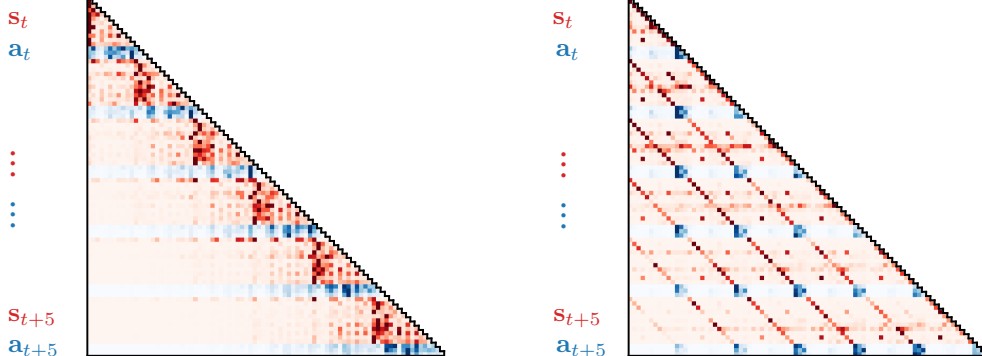

**Figure 4 (Attention patterns)** We observe two distinct types of attention masks during trajectory prediction. In the first, both states and actions are dependent primarily on the immediately preceding transition, corresponding to a model that has learned the Markov property. The second strategy has a striated appearance, with state dimensions depending most strongly on the same dimension of multiple previous timesteps. Surprisingly, actions depend more on past actions than they do on past states, reminiscent of the action smoothing used in some trajectory optimization algorithms (Nagabandi et al., 2019). The above masks are produced by a first- and third-layer attention head during sequence prediction on the hopper benchmark; reward dimensions are omitted for this visualization.[1]

## 4.2 Reinforcement Learning and Control

**Offline reinforcement learning.** We evaluate the Trajectory Transformer on a number of environments from the D4RL offline benchmark suite (Fu et al., 2020), including the locomotion and AntMaze domains. This evaluation is the most difficult of our control settings, as reward-maximizing behavior is the most qualitatively dissimilar from the types of behavior that are normally associated with unsupervised modeling – namely, imitative behavior. Results for the locomotion environments are shown in Table 1. We compare against five other methods spanning other approaches to data-driven control: (1) behavior-regularized actor-critic (BRAC; Wu et al. 2019) and conservative $Q$-learning (CQL; Kumar et al. 2020a) represent the current state-of-the-art in model-free offline RL; model-based offline planning (MBOP; Argenson & Dulac-Arnold 2021) is the best-performing prior offline trajectory optimization technique; decision transformer (DT; Chen et al. (2021)) is a concurrently-developed sequence-modeling approach that uses return-conditioning instead of planning; and behavior-cloning (BC) provides the performance of a pure imitative method.

The Trajectory Transformer performs on par with or better than all prior methods (Table 1). The two discretization variants of the Trajectory Transformer, uniform and quantile, perform similarly on all environments except for HalfCheetah-Medium-Expert, where the large range of the velocities prevents the uniform discretization scheme from recovering the precise actuation required for enacting the expert policy. As a result, the quantile discretization approach achieves a return of more than twice that of the uniform discretization.

**Combining with $Q$-functions.** Though Monte Carlo value estimates are sufficient for many standard offline RL benchmarks, in sparse-reward and long-horizon settings they become too uninformative to guide the beam-search-based planning procedure. In these problems, the value estimate from the Transformer can be replaced with a $Q$-function trained via dynamic programming. We explore this combination by using the $Q$-function from the implicit $Q$-learning algorithm (IQL; Kostrikov et al. 2021) on the AntMaze navigation tasks (Fu et al., 2020), for which there is only a sparse reward upon reaching the goal state. These tasks evaluate temporal compositionality because they require stitching together multiple zero-reward trajectories in the dataset to reach a designated goal.

AntMaze results are provided in Table 2. $Q$-guided Trajectory Transformer planning outperforms all prior methods on all maze sizes and dataset compositions. In particular, it outperforms the IQL method from which we obtain the $Q$-function, underscoring that planning with a $Q$-function as a

---

[1]More attention visualizations can be found at `trajectory-transformer.github.io/attention`

| Dataset | Environment | BC | MBOP | BRAC | CQL | DT | TT (uniform) | TT (quantile) |
|---------|-------------|-----|------|------|-----|-----|--------------|---------------|
| Med-Expert | HalfCheetah | 59.9 | 105.9 | 41.9 | 91.6 | 86.8 | 40.8 ±2.3 | 95.0 ±0.2 |
| Med-Expert | Hopper | 79.6 | 55.1 | 0.9 | 105.4 | 107.6 | 106.0 ±0.28 | 110.0 ±2.7 |
| Med-Expert | Walker2d | 36.6 | 70.2 | 81.6 | 108.8 | 108.1 | 91.0 ±2.8 | 101.9 ±6.8 |
| Medium | HalfCheetah | 43.1 | 44.6 | 46.3 | 44.0 | 42.6 | 44.0 ±0.31 | 46.9 ±0.4 |
| Medium | Hopper | 63.9 | 48.8 | 31.3 | 58.5 | 67.6 | 67.4 ±2.9 | 61.1 ±3.6 |
| Medium | Walker2d | 77.3 | 41.0 | 81.1 | 72.5 | 74.0 | 81.3 ±2.1 | 79.0 ±2.8 |
| Med-Replay | HalfCheetah | 4.3 | 42.3 | 47.7 | 45.5 | 36.6 | 44.1 ±0.9 | 41.9 ±2.5 |
| Med-Replay | Hopper | 27.6 | 12.4 | 0.6 | 95.0 | 82.7 | 99.4 ±3.2 | 91.5 ±3.6 |
| Med-Replay | Walker2d | 36.9 | 9.7 | 0.9 | 77.2 | 66.6 | 79.4 ±3.3 | 82.6 ±6.9 |
| **Average** | | 47.7 | 47.8 | 36.9 | 77.6 | 74.7 | 72.6 | 78.9 |

**Table 1 (Offline reinforcement learning)** The Trajectory Transformer (TT) performs on par with or better than the best prior offline reinforcement learning algorithms on D4RL locomotion (v2) tasks. Results for TT variants correspond to the mean and standard error over 15 random seeds (5 independently trained Transformers and 3 trajectories per Transformer). We detail the sources of the performance for other methods in Appendix **??**.

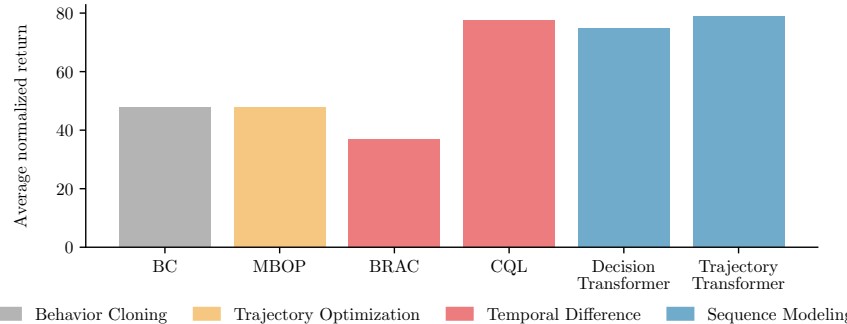

**Figure 5 (Offline averages)** A plot showing the average per-algorithm performance in Table 1, with bars colored according to a crude algorithm categorization. In this plot, "Trajectory Transformer" refers to the quantile discreization variant.

search heuristic can be less susceptible to errors in the $Q$-function than policy extraction. However, because the $Q$-guided planning procedure still benefits from the temporal compositionality of both dynamic programming and planning, it outperforms return-conditioning approaches, such as the Decision Transformer, that suffer due to the lack of complete demonstrations in the AntMaze datasets.

**Imitation and goal-reaching.** We additionally plan with the Trajectory Transformer using standard likelihood-maximizing beam search, as opposed to the return-maximizing version used for offline RL. We find that after training the model on datasets collected by expert policies (Fu et al., 2020), using beam search as a receding-horizon controller achieves an average normalized return of $104\%$ and $109\%$ in the Hopper and Walker2d environments, respectively, using the same evaluation protocol of 15 runs described as in the offline RL results. While this result is perhaps unsurprising, as behavior cloning with standard feedforward architectures is already able to reproduce the behavior of the expert policies, it demonstrates that a decoding algorithm used for language modeling can be effectively repurposed for control.

Finally, we evaluate the goal-reaching variant of beam-search, which conditions on a future desired state alongside previously encountered states. We use a continuous variant of the classic four rooms environment as a testbed (Sutton et al., 1999). Our training data consists of trajectories collected by a pretrained goal-reaching agent, with start and goal states sampled uniformly at random across the state space. Figure 6 depicts routes taken by the the planner. Anti-causal conditioning on a future state allows for beam search to be used as a goal-reaching method. No reward shaping, or rewards of any sort, are required; the planning method relies entirely on goal relabeling. An extension of this experiment to procedurally-generated maps is described in Appendix **??**.

| Dataset | Environment | BC | CQL | IQL | DT | TT $(+Q)$ |
|---------|-------------|-----|-----|-----|-----|-----------|
| Umaze | AntMaze | 54.6 | 74.0 | 87.5 | 59.2 | 100.0 $\pm 0.0$ |
| Medium-Play | AntMaze | 0.0 | 61.2 | 71.2 | 0.0 | 93.3 $\pm 6.4$ |
| Medium-Diverse | AntMaze | 0.0 | 53.7 | 70.0 | 0.0 | 100.0 $\pm 0.0$ |
| Large-Play | AntMaze | 0.0 | 15.8 | 39.6 | 0.0 | 66.7 $\pm 12.2$ |
| Large-Diverse | AntMaze | 0.0 | 14.9 | 47.5 | 0.0 | 60.0 $\pm 12.7$ |
| Average | | 10.9 | 44.9 | 63.2 | 11.8 | 84.0 |

**Table 2 (Combining with $Q$-functions)** Performance on the sparse-reward AntMaze (v0) navigation task. Using a $Q$-function as a search heuristic with the Trajectory Transformer (TT $(+Q)$) outperforms policy extraction from the $Q$-function (IQL) and return-conditioning approaches like the Decision Transformer (DT). We report means and standard error over 15 random seeds for TT $(+Q)$; baseline results are taken from Kostrikov et al. (2021).

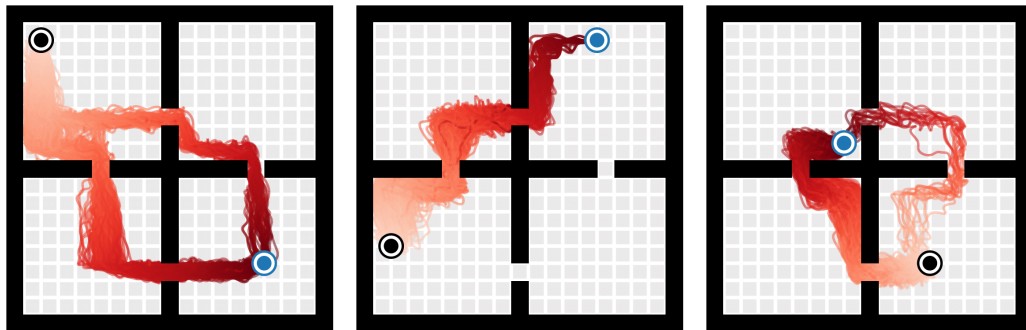

**Figure 6 (Goal-reaching)** Trajectories collected by TTO with anti-causal goal-state conditioning in a continuous variant of the four rooms environment. Trajectories are visualized as curves passing through all encountered states, with color becoming more saturated as time progresses. Note that these curves depict real trajectories collected by the controller and not sampled sequences. The starting state is depicted by ◉ and the goal state by ◉. Best viewed in color.

## 5  Discussion and Limitations

We have presented a sequence modeling view on reinforcement learning that enables us to derive a single algorithm for a diverse range of problem settings, unifying many of the standard components of reinforcement learning algorithms (such as policies, models, and value functions) under a single sequence model. The algorithm involves training a sequence model jointly on states, actions, and rewards and sampling from it using a minimally modified beam search. Despite drawing from the tools of large-scale language modeling instead of those normally associated with control, we find that this approach is effective in imitation learning, goal-reaching, and offline reinforcement learning.

However, prediction with Transformers is currently slower and more resource-intensive than prediction with the types of single-step models often used in model-based control, requiring up to multiple seconds for action selection when the context window grows too large. This precludes real-time control with standard Transformers for most dynamical systems. While the beam-search-based planner is conceptually an instance of model-predictive control, and as such could be applicable wherever model-based RL is, in practice the slow planning also makes online RL experiments unwieldy. (Computationally-efficient Transformer architectures (Tay et al., 2021) could potentially cut runtimes down substantially.) Further, we have chosen to discretize continuous data to fit a standard architecture instead of modifying the architecture to handle continuous inputs. While we found this design to be much more effective than conventional continuous dynamics models, it does in principle impose an upper bound on prediction precision.

This paper is an investigation of a minimal type of algorithm that can be applied to RL problems. While one of the interesting implications of our results is that RL problems can be reframed as supervised learning tasks with an appropriate choice of model, the most practical instantiation of this idea may come from combinations with dynamic programming techniques, as suggested by the effectiveness of the Trajectory Transformer with $Q$-guided planning.

**Code References**

We used the following open-source libraries for this work: NumPy (Harris et al., 2020), PyTorch (Paszke et al., 2019), and minGPT (Karpathy, 2020).

**Acknowledgements**

We thank Ethan Perez and Max Kleiman-Weiner for helpful discussions and Ben Eysenbach for feedback on an early draft. M.J. thanks Karthik Narasimhan for early inspiration about parallels between language modeling and model-based reinforcement learning. This work was partially supported by computational resource donations from Microsoft. M.J. is supported by fellowships from the National Science Foundation and the Open Philanthropy Project.

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
