# Appendix A   Model and Training Specification

**Architecture and optimization details.**   In all environments, we use a Transformer architecture with four layers and four self-attention heads. The total input vocabulary of the model is $V \times (N + M + 2)$ to account for states, actions, rewards, and rewards-to-go, but the output linear layer produces logits only over a vocabulary of size $V$; output tokens can be interpreted unambiguously because their offset is uniquely determined by that of the previous input. The dimension of each token embedding is 128. Dropout is applied at the end of each block with probability 0.1.

We follow the learning rate scheduling of (Radford et al., 2018), increasing linearly from 0 to $2.5 \times 10^{-4}$ over the course of 2000 updates. We use a batch size of 256.

**Hardware.**   Model training took place on NVIDIA Tesla V100 GPUs (NCv3 instances on Microsoft Azure) for 80 epochs, taking approximately 6-12 hours (varying with dataset size) per model on one GPU.

# Appendix B   Discrete Oracle

The discrete oracle in Figure 3 is the maximum log-likelihood attainable by a model under the uniform discretization granularity. For a single state dimension $i$, this maximum is achieved by a model that places all probability mass on the correct token, corresponding to a uniform distribution over an interval of size

$$\frac{r_i - \ell_i}{V}.$$

The total log-likelihood over the entire state is then given by:

$$\sum_{i=1}^{N} \log \frac{V}{r_i - \ell_i}.$$

# Appendix C   Baseline performance sources

**Offline reinforcement learning**   The results for CQL, IQL, and DT are from Table 1 in Kostrikov et al. (2021). The results for MBOP are from Table 1 in Argenson & Dulac-Arnold (2021). The results for BRAC are from Table 2 in Fu et al. (2020). The results for BC are from Table 1 in Kumar et al. (2020a).

# Appendix D   Datasets

The D4RL dataset (Fu et al., 2020) used in our experiments is under the Creative Commons Attribution 4.0 License (CC BY). The license information can be found at

> `https://github.com/rail-berkeley/d4rl/blob/master/README.md`

under the "Licenses" section.

# Appendix E    Beam Search Hyperparameters

| **Beam width** | maximum number of hypotheses retained during beam search | 256 |
|---|---|---|
| **Planning horizon** | number of transitions predicted by the model during | 15 |
| **Vocabulary size** | number of bins used for autoregressive discretization | 100 |
| **Context size** | number of input $(\mathbf{s}_t, \mathbf{a}_t, \mathbf{r}_t, R_t)$ transitions | 5 |
| $k_{\text{obs}}$ | top-$k$ tokens from which observations are sampled | 1 |
| $k_{\text{act}}$ | top-$k$ tokens from which actions | 20 |

Beam width and context size are standard hyperparameters for decoding Transformer language models. Planning horizon is a standard trajectory optimization hyperparameter. The hyperparameters $k_{\text{obs}}$ and $k_{\text{act}}$ indicate that actions are sampled from the most likely $20\%$ of action tokens and next observations are decoded greedily conditioned on previous observations and actions.

In many environments, the beam width and horizon may be reduced to speed up planning without affecting performance. Examples of these configurations are provided in the reference implementation: github.com/jannerm/trajectory-transformer.

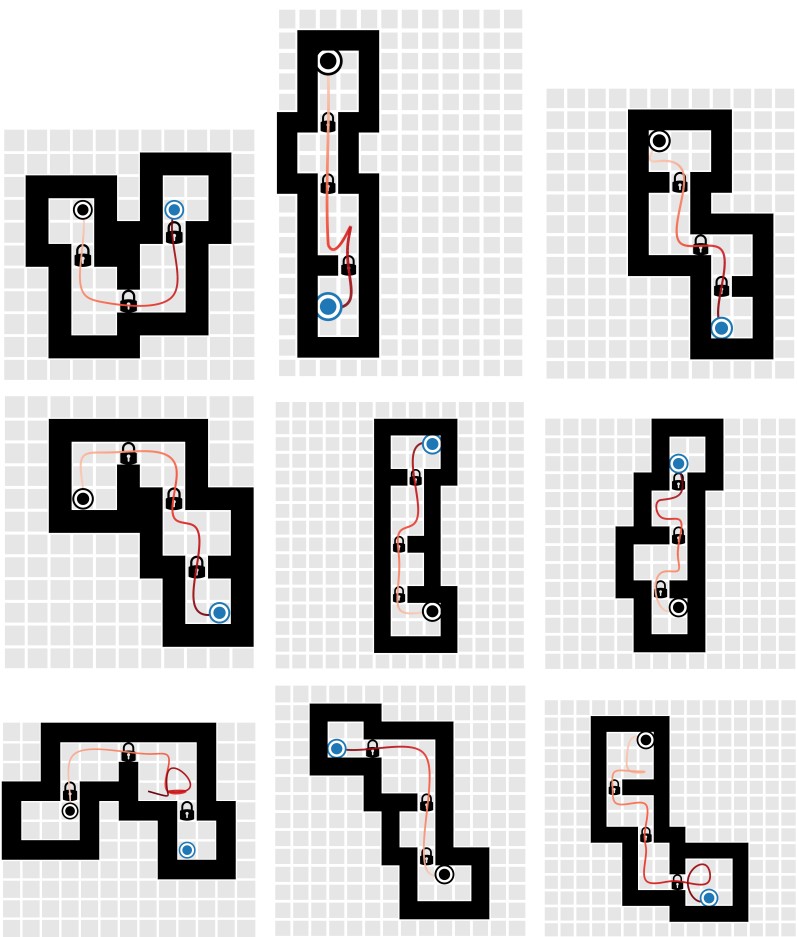

**Figure 7 (Goal-Reaching in MiniGrid)** Example paths of the Trajectory Transformer planner in the `MiniGrid-MultiRoom-N4-S5`. Lock symbols indicate doors.

## Appendix F  Goal-Reaching on Procedurally-Generated Maps

The method evaluated here and the experimental setup is identical to that described in Section 3.2 (Goal-conditioned reinforcement learning), with one distinction: because the map changes each episode, the Transformer model has an additional context embedding that is a function of the current observation image. This embedding is the output of a small convolutional neural network and is added to the token embeddings analogously to the treatment of position embeddings. The agent position and goal state are not included in the map; these are provided as input tokens as described in Section 3.2.

The action space of this environment is discrete. There are seven actions, but only four are required to complete the tasks: turning left, turning right, moving forward, and opening a door. The training data is a mixture of trajectories from a pre-trained goal-reaching policy and a uniform random policy.

94% of testing goals are reached by the model on held-out maps. Example paths are shown in Figure 7.