# OpenReview forum: "Offline Reinforcement Learning as One Big Sequence Modeling Problem"
_NeurIPS.cc/2021/Conference — NeurIPS 2021 Spotlight_

### Official Review · Reviewer_Ldp2 · 2021-07-16

**Rating:** 7
**Confidence:** 3

**Summary:**

The authors propose a sequence modeling approach for reinforcement learning (RL) problems. This proposed approaches enables to tackle diverse range of problem settings and unifies many of the standard components RL algorithms (e.g. as policies, models, and value functions) under a single sequence model. The authors present empirical results that illustrate the effectiveness of their model on imitation learning, goal-reaching, and offline reinforcement learning problems.

**Limitations And Societal Impact:**

Yes. Several limitations of the work are described in the Discussion section. There are no ethical concerns, as far as I'm concerned.

See above for suggestions.



**Main Review:**

*Originality*: In my view, the usage of contemporary sequence modeling toolbox as a for sequential decision-making tasks is sufficiently novel. It provides new prospective on how to make use of highly successful transformer archtecture in various contexts and get good results. The related work adequately cited?

*Quality*: The submission is technically sound. Claims are supported by the large number of experimental results. The visualisations and qualitative examples are also assisting the reader.

*Clarity*: The manuscript is clear, well organized and easy to follow.

*Significance*: In my view, these are interesting results that the community can learn from and build upon.

### Cons

- The authors admit that predictions with Transformers is slower and more resource intensive. This is expected. However, I think the reader would benefit from knowing how much slower TTO is compared to baselines. Hence I ask the authors to add that information for their experiments (imitation learning, goal-conditioned RL, and offline RL).
- I'd be curious to see if goal-reaching is possible in more challenging domains compared to 4 rooms. Perhaps in procedurally generated mazes that change at every episode (e.g. MiniGrid).

# Post rebuttal update

I thanks the authors for their response and for running additional experiments using MiniGrid. I've also read the reviews by my fellow reviewers and the authors rebuttal to them (including many additional experiments). Therefore, I have increased my score by 1.

**Time Spent Reviewing:**

4

---

> ### Author Response · Authors · 2021-08-10
> **Response to Reviewer Ldp2**
>
> Thank you for taking the time to review our work.\
> &nbsp;
>
> > _The authors admit that predictions with Transformers is slower and more resource intensive. This is expected. However, I think the reader would benefit from knowing how much slower TTO is compared to baselines._
>
> We have included time profiles of our trajectory optimizer for various horizons and context window sizes at the following webpage:\
> [anonymized-transformer.github.io/benchmarking](https://anonymized-transformer.github.io/benchmarking/) \
> &nbsp;
>
> > _I'd be curious to see if goal-reaching is possible in more challenging domains compared to 4 rooms. Perhaps in procedurally generated mazes that change at every episode (e.g. MiniGrid)._
>
> Thank you for the suggestion. We have run another goal-reaching experiment in the procedurally-generated MiniGrid MultiRoom environment and included the results here:\
> [anonymized-transformer.github.io/minigrid](https://anonymized-transformer.github.io/minigrid/)\
> The main result is that the trajectory optimizer reaches 94% of goals on new test maps, but we have included more information about the setup and lots of qualitative results at the above link. We will add this experiment to the paper.\
> &nbsp;
>
> Please let us know if you have any further concerns about the paper.

---

> > ### Comment · Reviewer_Ldp2 · 2021-08-16
> > **Response to author comments**
> >
> > I thanks the authors for their response and for running additional experiments using MiniGrid. I've also read the reviews by my fellow reviewers and the authors rebuttal to them (including many additional experiments). Therefore, I have increased my score by 1.

---

### Official Review · Reviewer_6knX · 2021-07-17

**Rating:** 7
**Confidence:** 4

**Summary:**

In a departure from traditional RL approaches, the paper applies transformer-based sequence modeling (with beam search) to a variety of RL tasks. Evaluations show the proposed Trajectory Transformer (TTO) to be competitive with baselines on three domains.

**Limitations And Societal Impact:**

yes

**Main Review:**

- Originality

The proposed approach is novel and well-motivated, given the successes of transformers in sequence modeling.


- Quality

The experimental results indicate that TTO can be usefully applied to at least a few, somewhat peripheral (non-online) tasks that fall within the broad collection of work associated with RL. However, the paper is worded in a way that is easy to interpret too broadly, starting with the catchy title, which could make many readers think of online RL, since that is the core of the field. Some readers may not even realize that online RL is excluded by the final sentence of the abstract, since imitation learning and offline RL are common pretraining steps performed prior to online RL training, and since goal conditioning often appears in the online RL setting. Expectations could be better managed by making the title less all-inclusive. For instance, “**Non-Online** Reinforcement Learning as One Big Sequence Modeling Problem”. (I’m sure the authors could think of something better than I just did.) And line 308 in the conclusion could also be more carefully qualified, as “**certain** reinforcement learning problems”.

Any reader who initially assumes that this work reduces online RL to sequence modeling, freeing them from the tyranny of estimating cumulative reward, will be disappointed by the fine print that follows. As explained in section 3.2, this paper’s imitation learning task is not of the usual kind, and the goal-conditioned task is outside the online setting. Line 86 says “Our model treats states, actions, and rewards interchangeably”, but this is not quite true for the offline RL task, where reward-to-go is used as the beam search heuristic. Line 193 argues that these reward estimates need not be particularly accurate, but why assume that online RL requires more accurate value estimates than beam search does?

I don’t believe that a bold new RL approach such as sequence modeling needs to achieve SOTA or even be applicable to the most common RL problems in order to merit publication. But I do believe the claims should be more explicitly scoped.


- Clarity

Apart from needing greater clarity in its claims, the paper is generally clear and well-written. I see a few other areas for improvement.

Line 107 says that continuous states and actions were discretized “for processing by a discrete-token architecture”, leaving readers to assume that the transformer architecture itself is being discussed. But transformers have already been successfully applied to online RL in a variety of ways, using both continuous and discrete observations and/or actions. So it would help to explain more directly that the discrete-token architecture chosen for this work was minGPT along with its discrete token embedding and decoding machinery.

This work replaces some of the traditional RL machinery with beam search, which is a tricky tool with its own practical limitations that deserve much greater attention for the benefit of potential users in the RL community. Algorithm 1 describes beam search at a certain level, but the algorithm box or accompanying text should include basic explanations of the individual symbols and how they are to be used in an RL task. In addition, the critical issues like beam size, branching factor, search depth, and heuristic pruning factors need to be given concrete numbers in the discussion of each task. Many of these details could be relegated to an appendix, but they are as important as any hyperparameter value.

What is the interpretation of the vertical axis in Figure 4 and Figure 6?

The binary footprint of Figure 5 should be compressed. It dramatically slows down my browser every time I try to scroll past it.


- Significance

This is an interesting new approach that may deliver benefits in certain types of RL problems, and may even lead to useful online techniques.


- Conclusion

Given the shortcomings described above, I cannot support publication of the paper in its current form. But I believe the work makes important contributions which deserve a wide audience, so I hope the paper’s claims can be clarified so as not to mislead over-anxious readers.

- POST-RESPONSES UPDATE

The authors have addressed all of my reservations. With these revisions, the paper seems properly scoped and easier to understand. I have raised my score by 2 points.

**Time Spent Reviewing:**

12

---

> ### Author Response · Authors · 2021-08-10
> **Response to Reviewer 6knX**
>
> Thank you for your review of our work. We begin by responding to your main concern about scoping and online RL:
>
> > _However, the paper is worded in a way that is easy to interpret too broadly, starting with the catchy title, which could make many readers think of online RL, since that is the core of the field. Some readers may not even realize that online RL is excluded by the final sentence of the abstract, since imitation learning and offline RL are common pretraining steps performed prior to online RL training, and since goal conditioning often appears in the online RL setting._
>
> We do want to ensure that we scope the discussion of the paper appropriately, so will add “Offline” to the title to emphasize that we focus on the offline setting in our experiments. We will also make this more explicit in the conclusion. Please let us know if there are other specific places where you feel that the claims overreach.
>
> That said, we also want to clarify that TTO is effectively a model-based planning algorithm (with a high-capacity long-horizon model), so in principle it is just as suitable for online RL as any other MPC-based planner. Running TTO online would simply use the data-collection loop standard in model-based RL algorithms [[1](https://arxiv.org/abs/1805.12114), [2](https://arxiv.org/abs/1909.11652)], in which the planning procedure is run to collect more data and the newly collected data further improves the model. We focus on offline RL in our evaluation because the sequence modeling approach to decision-making has the most potential in data-rich settings, and this is well-matched to offline RL because of the emphasis on large static datasets. However, while we would not expect there to be a large benefit of using Transformers in the low-data online regime, online RL is not conceptually excluded by Transformers. \
> &nbsp;
>
> We now respond to the itemized questions:
>
> > _As explained in section 3.2, this paper’s imitation learning task is not of the usual kind, and the goal-conditioned task is outside the online setting._
>
> Could you clarify what you mean by “not of the usual kind” of imitation learning? While our imitation learning _method_ differs from the standard formulation, we use it in exactly the same problem setting as standard imitative methods and compare to conventional behavior cloning. \
> As for goal-reaching, the same holds as in our first response: we focus on settings where sequence models can make use of large datasets, but the trajectory optimizer does not exclude the data recollection and continued model training of online RL. \
> &nbsp;
>
> > _Line 193 argues that these reward estimates need not be particularly accurate, but why assume that online RL requires more accurate value estimates than beam search does?_
>
> Thank you, we will clarify this point. When we say that “our method does not require the estimated values to be particularly accurate”, we were not comparing to online RL algorithms. Instead, we were comparing a method that uses values as a search heuristic (TTO) to one that acts greedily with respect to values (anything with policy extraction). The extreme case might illustrate the reason: if our beam width were large enough to decode all possible N-length trajectories, we would not need the value estimate at all! The value estimate just allows us to have a smaller beam width by biasing the search to promising candidates.
>
> As a concrete example: if we optimize greedily with respect to the reward-to-go predictions on the hopper-medium-replay task (effectively a horizon of 1), TTO gets an average normalized return of 35.0. The full method, which only uses reward-to-go predictions as a search heuristic for planning, gets an average normalized score of 99.4. \
> &nbsp;
>
> > _This work replaces some of the traditional RL machinery with beam search, which is a tricky tool with its own practical limitations that deserve much greater attention for the benefit of potential users in the RL community._
>
> We have included hyperparameters here: \
> [anonymized-transformer.github.io/hyperparameters](https://anonymized-transformer.github.io/hyperparameters/) \
> This table will be included in the appendix. \
> &nbsp;
>
> > _What is the interpretation of the vertical axis in Figure 4 and Figure 6?_
>
> Normalized returns on the D4RL benchmark datasets. We have added this in the figure: \
> [anonymized-transformer.github.io/offline_rl](https://anonymized-transformer.github.io/offline_rl/) \
> More information about the normalized scoring can be found in the [D4RL paper](https://arxiv.org/abs/2004.07219) (Table 2 in the appendix in particular). \
> &nbsp;
>
> > _Line 107 says that continuous states and actions were discretized “for processing by a discrete-token architecture”, leaving readers to assume that the transformer architecture itself is being discussed..._
>
> We will specify that we are referring to the (min)GPT 2 architecture [[1](https://cdn.openai.com/better-language-models/language_models_are_unsupervised_multitask_learners.pdf), [2](https://github.com/karpathy/minGPT)]. \
> &nbsp;
>
> > _Line 86 says “Our model treats states, actions, and rewards interchangeably”, but this is not quite true for the offline RL task, where reward-to-go is used as the beam search heuristic._
>
> Thank you for pointing this out; we have removed this word. \
> &nbsp;
>
> > _The binary footprint of Figure 5 should be compressed. It dramatically slows down my browser every time I try to scroll past it._
>
> Thank you for catching this. We were able to compress this figure 50x.

---

> > ### Comment · Reviewer_6knX · 2021-08-21
> > **Algorithm 1**
> >
> > Thank you for addressing most of my concerns!
> >
> > The only item for which I didn't see a response is the missing descriptions of symbols in Algorithm 1. Beam search is central to your proposal, so readers unfamiliar with beam search should be walked through the details. And there should be clear explanations relating *planning horizon* and *context size* from your new table in the appendix to the symbols in Algorithm 1.

---

> > > ### Author Response · Authors · 2021-08-23
> > > **Algorithm 1**
> > >
> > > Thank you for the comment! We have streamlined the presentation of beam search and provided an expanded description at the following webpage:\
> > > [anonymized-transformer.github.io/beam-search](https://anonymized-transformer.github.io/beam-search/) \
> > > The description contains details about how parameters of the trajectory optimizer, such as context size and planning horizon (in terms of transitions), translate into parameters for the generic beam search algorithm (in terms of tokens). It also defines all symbols appearing in the pseudocode.
> > >
> > > We will include this expanded description in Section 3.2 of the paper.

---

> > > > ### Comment · Reviewer_6knX · 2021-08-24
> > > > **Algo 1**
> > > >
> > > > Very nicely written, it's a pleasure to read!
> > > >
> > > > Algo 1 seems to use the same symbol y to signify two different things:  one vocabulary token, and one T-length sequence from the final hypothesis set.

---

> > > > > ### Author Response · Authors · 2021-08-24
> > > > > **Algorithm 1**
> > > > >
> > > > > Nice catch! We indeed missed a `\mathbf` in L5 to denote that the $\mathbf{y}$ there represents a sequence. (It's now updated.)
> > > > >
> > > > > Many thanks for reading it closely enough to spot that!

---

> ### Author Response · Authors · 2021-08-20
> **Follow-up**
>
> Hi Reviewer 6knX, do the proposed changes sufficiently address the scoping concern? To summarize:
> - We will add "Offline" to the title.
> - In the introduction and conclusion, we will make explicit that we evaluate the method in settings where offline data is available for training the Transformer model.
> - Also in the conclusion, we will discuss the relation of Transformer trajectory optimization (TTO) to standard trajectory optimization, and what would be required to deploy this trajectory optimizer in the online setting.
>
> Do you have any other unaddressed concerns?

---

### Official Review · Reviewer_oBoc · 2021-07-27

**Rating:** 5
**Confidence:** 5

**Summary:**

This paper attempts to reformulate RL as a sequence modeling problem and hence exploit the recent advances in sequence modeling like Transformers to design an RL agent. Authors show that with little modification to existing sequence to sequence models and beam search techniques, they can perform well in imitation learning, goal-conditioned RL, and offline RL.

**Ethical Concerns:**

None.

**Limitations And Societal Impact:**

Check the main review.


**Main Review:**

While this is an interesting research direction, I have several major concerns about this work.

1. I do not think that throwing away all the dynamic programming techniques and replacing them with Transformers will help us solve RL. While I disagree with this idea, my decision is not based on the fact that I do not like this approach.
2. I feel like authors do not highlight the limitations of this simple sequence to sequence approach well enough. Limitations mentioned in section 5 are not serious limitations! Would this approach work when there is stochasticity? Non-stationarity? More diverse tasks? This simple approach would suffer a lot in complex environments and this has to be made clear. While it is ok for having these limitations, it is crucial for this paper to have an explicit limitations section so that it does not mislead the community.
3. Why did the authors decide to go with the proposed discretization approach? It would be nice to see the performance of some baseline representations to understand the significance of this representation.
4. Line 133 -> The maximum tokens in 512. What are the values of N and M? How many transitions can you keep in the history if your maximum token length is 512? This is very useful information to know.
5. Again, Figure-1 experiments are very misleading. You are comparing Transformers models with PETS. PETS uses simple feedforward networks. I would like to see the comparison between Transformers and PlaNet (Hafner et al 2019).
6. Figure 3 - Are these the only two types of attention you have seen in the trained model? How much cherry-picking happened here? Please explain the selection process here.
7. How is this work different from [2]? It is not clear whether the performance trends that authors show here would be the same if they use proper recurrent models in their baselines as well.


Minor comments:

1. Figure 2 and Figure 4 - Please specify the number of runs.
2. Figure 4 - What is the y axis?

References:

[1] Learning Latent Dynamics for Planning from Pixels. Hafner et al 2019
[2] Reinforcement Learning Upside Down: Don't Predict Rewards -- Just Map Them to Actions. Schmidhuber 2019


**Time Spent Reviewing:**

5

---

> ### Author Response · Authors · 2021-08-10
> **Response to Reviewer oBoc**
>
> Thank you for taking the time to review our work. We have organized our response with one blockquote per question, except for your first two questions, which we have grouped together.\
> &nbsp;
>
> > _I do not think that throwing away all the dynamic programming techniques and replacing them with Transformers will help us solve RL… This simple approach would suffer a lot in complex environments and this has to be made clear._
>
> Our conclusion is not that we should throw away dynamic programming techniques, and we do not claim this. (We will clarify this in the paper more explicitly.) Rather, we make the perhaps surprising observation that many settings encompassed by the RL problem can be tackled with a trajectory optimizer that is almost indistinguishable from sequence modeling. That does not mean this algorithm is uniformly better than dynamic programming, especially in the types of complex settings where we do not yet have mature benchmarks for evaluating this type of claim, but the evidence we have for this algorithm being viable is a direct comparison against contemporary RL algorithms in problem settings that are receiving some of the most current focus.
>
> Indeed, in future work, it's likely such an approach could be combined with dynamic programming -- it is not mutually exclusive. However, we believe it is valuable to study it in isolation to understand the minimal approach that can address RL problems.\
> &nbsp;
>
> > _Are these the only two types of attention you have seen in the trained model? How much cherry-picking happened here?_
>
> We have included 1000 randomly-selected attention maps on the following webpage:\
> [anonymized-transformer.github.io/attention](https://anonymized-transformer.github.io/attention/) \
> &nbsp;
>
> > _Why did the authors decide to go with the proposed discretization approach?_
>
> This question can be interpreted in two different ways, so we include answers to both here: \
> **a) Why discretize the states at all?** We also tested a continuous variant of the transformer architecture with outputs parameterizing a Gaussian distribution, but found that this did not work substantially better than the single-step MLP baseline: \
> [anonymized-transformer.github.io/continuous](https://anonymized-transformer.github.io/continuous/) \
> One of the biggest drawbacks of this formulation is that it does not allow for multimodal prediction like the discretized approach does. We tried to retain the benefits of multimodality with continuous states by parameterizing GMM predictions with the transformer, but found this to be unstable and worse than unimodal Gaussian predictions.
>
> **b) Why use this particular naïve discretization technique?** In short, because it was simple and it worked. We found that more sophisticated discretization schemes like quantile regression improved performance on some tasks where precise actuation is important (like HalfCheetah): \
> [anonymized-transformer.github.io/quantile_regression](https://anonymized-transformer.github.io/quantile_regression/) \
> but the uniform bucketing approach had the best performance-to-complexity ratio. The implementation of this discretization technique, plus a small visualization of how it differs from the naïve one, is provided here: \
> [anonymized-transformer.github.io/quantile_implementation](https://anonymized-transformer.github.io/quantile_implementation/) \
> We will include the results from the more sophisticated discretization scheme in the paper for comparison.\
> &nbsp;
>
> > _Again, Figure-1 experiments are very misleading. You are comparing Transformers models with PETS. PETS uses simple feedforward networks. I would like to see the comparison between Transformers and PlaNet (Hafner et al 2019)._
>
> We have run the PlaNet model on the same Humanoid prediction task, for direct comparison to the Transformer predictions in Figure 1: \
> [anonymized-transformer.github.io/planet](https://anonymized-transformer.github.io/planet/)
>
> Two quick notes for interpreting these results: \
> **a)** The planning horizon used by PlaNet is 12, much shorter than the prediction horizon shown here. We found that the PlaNet model is indeed accurate on the order of a dozen steps, consistent with the original reporting. \
> **b)** The PlaNet architecture was originally designed for image observations, so we had to make a few minor modifications (for example, to the observation model). Wherever possible, we left design decisions (both in terms of the architecture and the training procedure) intact.
>
> We found that PlaNet does not perform better than the single-step baseline. This is expected, as the dynamics of this task are Markovian, meaning that the improved performance of the Transformer is due to architectural considerations besides the longer context window (See lines 237-244 of the submission for further discussion.) \
> &nbsp;
>
> > _How is this work different from [2]?_
>
> UDRL [[2]](https://arxiv.org/abs/1912.02875) conditions a policy on a desired return and samples a single action given that return. We do not do return-conditioning, but instead sample plausible trajectories (consisting of states, actions, rewards, and returns) and select that which maximizes expected cumulative reward. Our approach is more similar to a trajectory optimization method (hence the name “Transformer trajectory optimization”) than it is to a return-conditioned policy approach.
>
> It is worth mentioning that there is a concurrent paper with a similar-sounding name (“Decision Transformer: Reinforcement Learning via Sequence Modeling”) that does condition on desired returns as in [[2]](https://arxiv.org/abs/1912.02875), but this technique is not a part of our method. \
> &nbsp;
>
> > _What are the values of N and M? How many transitions can you keep in the history if your maximum token length is 512?_
>
> The dimensionalities of the benchmark environments are: \
> **a)** Hopper: N=11, M=3, max sequence length with a 512-token window=512/(11+3+2)=32 \
> **b)** Walker2d: N=17, M=6, max sequence length=20 \
> **c)** HalfCheetah: N=17, M=6, max sequence length=20 \
> **d)** Humanoid: N=46, M=17, max sequence length=7 \
> The +2 in the max sequence calculation comes from addition of the reward and reward-to-go.
>
>
> For tasks with larger state spaces (like Humanoid), if a larger context were needed we could replace the Transformer (with attention complexity quadratic in sequence length) with a more efficient architecture like the [Reformer](https://arxiv.org/abs/2001.04451) ($\mathcal{O}(n \log n)$ complexity) or [Performer](https://arxiv.org/abs/2009.14794) ($\mathcal{O}(n)$ complexity). We expect that this will be useful in some experimental settings, but we did not find it to be required in the standard benchmark tasks considered in this paper, so we leave the extension to longer-horizon conditioning to future work. (We found that we could use smaller context windows than the maximum allowed by GPU memory without suffering from performance drops; see [hyperparameters](https://anonymized-transformer.github.io/hyperparameters/).) \
> &nbsp;
>
> > _Figure 2 and Figure 4 - Please specify the number of runs._
>
> We sampled 32 trajectories from all models for the prediction result and used 15 runs (the same as the offline RL result in Figure 6) for the imitation result. We will add this information to the captions; thank you for catching its absence. \
> &nbsp;
>
> > _Figure 4 - What is the y axis?_
>
> Normalized returns on the D4RL benchmark datasets. We have added this in the figure: \
> [anonymized-transformer.github.io/offline_rl](https://anonymized-transformer.github.io/offline_rl/) \
> More information about the normalized scoring can be found in the [D4RL paper](https://arxiv.org/abs/2004.07219) (Table 2 in the appendix in particular). \
> &nbsp;
>
> Please let us know if you have any other concerns.

---

> ### Author Response · Authors · 2021-08-18
> **Author follow-up**
>
> Hi Reviewer oBoc, do you have any remaining concerns about the work?

---

> > ### Author Response · Authors · 2021-08-31
> > **Author follow-up**
> >
> > To summarize the main points:
> > 1. We have run the requested PlaNet baseline.
> > 2. We have provided 1000 more attention maps.
> > 3. We have benchmarked a continuous variant of the model and an alternative (quantile regression-based) discretization approach.
> > 4. We have clarified that we do not perform reward conditioning as in [[2]](https://arxiv.org/abs/1912.02875) but instead use a trajectory optimizer.
> >
> > (See the full response for answers to questions 1, 2, and 4.)
> >
> > Please let us know if you have any unaddressed questions.

---

### Author Response · Authors · 2021-08-10
**Consolidated Response**


We thank all reviewers for their time. We have responded individually to each reviewer, but summarize the main discussion points here:

1. **Cherry-picking:** There were concerns of cherry-picking the attention visualizations in Figure 3. We have included 1000 more randomly selected attention maps at the following webpage to alleviate these concerns: \
[anonymized-transformer.github.io/attention](https://anonymized-transformer.github.io/attention/)

2. **Procedurally generated goal-reaching environments:** We have run a goal-reaching experiment in the MiniGrid MultiRoom environment: \
[anonymized-transformer.github.io/minigrid](https://anonymized-transformer.github.io/minigrid/) \
The main result is that the trajectory optimizer reaches 94% of goals on unseen test maps. More information about the setup and qualitative results are provided in the above link.

3. **Recurrent model baseline:** We have run the PlaNet model [[Hafner, 2018]](https://arxiv.org/abs/1811.04551) on the Humanoid prediction task: \
[anonymized-transformer.github.io/planet](https://anonymized-transformer.github.io/planet/) \
We found that it did not perform better than the single-step model. This is expected because the dynamics are Markovian, meaning that the improved performance of the transformer is due to architectural considerations besides the longer context window (See lines 237-244 of the submission for further discussion.)

4. **Discretization:** We have shown that more sophisticated discretization techniques can improve performance on tasks where very precise actuation is important (like `halfcheetah-medium-expert`): \
[anonymized-transformer.github.io/quantile_regression](https://anonymized-transformer.github.io/quantile_regression/) \
though in other tasks the improved complexity does not yield improved performance. (We will include both in revisions of the paper.) \
The differences between these two implementations is detailed here: \
[anonymized-transformer.github.io/quantile_implementation](https://anonymized-transformer.github.io/quantile_implementation/). \
We also compared to a non-discretized version of the Transformer that replaces the tokenization with Gaussian heads: \
[anonymized-transformer.github.io/continuous](https://anonymized-transformer.github.io/continuous/)

5. **Scoping and suitability for online RL:** We will add “Offline” to the title to better communicate that our experiments use offline datasets. While we expect the sequence modeling approach to decision-making to have the most potential in data-rich settings, making it best-suited for the offline problem formulation, at its core the proposed approach is a trajectory optimizer, so it can be used in the standard model-based RL data recollection and improvement loop.

6. **Runtime information:** We have profiled the trajectory optimizer for various horizons and context window sizes: \
[anonymized-transformer.github.io/benchmarking](https://anonymized-transformer.github.io/benchmarking/)

7. **Relation to prior work:** We do not perform any desired return conditioning as described in Upside-Down RL [[Schmidhuber 2019]](https://arxiv.org/abs/1912.02877).

---

### Decision · Program_Chairs · 2021-09-27

**Decision:**

Accept (Spotlight)

**Comment:**

The reviews were thorough and there was good interaction with the authors.

I'd like to suggest that the authors may have come off as overly aggressive in their pushing for additional reviewer interaction, especially when the interaction was already quite good. This aggressiveness may backfire sometime in the future.

That said, the paper does seem to make a reasonable contribution and the delta between what is promised and the original submission is not overly large.